

# Identifying disease-associated signaling pathways through a novel effector gene analysis

Zhenshen Bao[1], Bing Zhang[1], Li Li[2], Qinyu Ge[1], Wanjun Gu[1] and Yunfei Bai[1]

[1] State Key Laboratory of Bioelectronics, School of Biological Sciences and Medical Engineering, Southeast University, Nanjing, Jiangsu, China
[2] Department of Respiratory Medicine, Zhongda Hospital, School of Medicine, Southeast University, Nanjing, Jiangsu, China

## ABSTRACT

**Background:** Signaling pathway analysis methods are commonly used to explain biological behaviors of disease cells. Effector genes typically decide functional attributes (associated with biological behaviors of disease cells) by abnormal signals they received. The signals that the effector genes receive can be quite different in normal vs. disease conditions. However, most of current signaling pathway analysis methods do not take these signal variations into consideration.

**Methods:** In this study, we developed a novel signaling pathway analysis method called signaling pathway functional attributes analysis (SPFA) method. This method analyzes the signal variations that effector genes received between two conditions (normal and disease) in different signaling pathways.

**Results:** We compared the SPFA method to seven other methods across 33 Gene Expression Omnibus datasets using three measurements: the median rank of target pathways, the median $p$-value of target pathways, and the percentages of significant pathways. The results confirmed that SPFA was the top-ranking method in terms of median rank of target pathways and the fourth best method in terms of median $p$-value of target pathways. SPFA's percentage of significant pathways was modest, indicating a good false positive rate and false negative rate. Overall, SPFA was comparable to the other methods. Our results also suggested that the signal variations calculated by SPFA could help identify abnormal functional attributes and parts of pathways. The SPFA R code and functions can be accessed at https://github.com/ZhenshenBao/SPFA.

# INTRODUCTION

Recently developed high-throughput functional genomics technologies have generated large amounts of experimental disease data and detected new biological information. Challenge for biologists is understanding the biological behaviors of disease cells using both newly generated disease data and existing biological knowledge. Signaling pathway analysis methods are used to better understand the biological behaviors of disease cells.

Corresponding author
Yunfei Bai, whitecf@seu.edu.cn

The understanding of biological behaviors of disease cells benefits to understand the pathological scenery and treatment. Over-representation analysis (ORA) based methods were initially presented as signaling pathway analysis methods to help biologists identify over-represented pathways from a list of relevant genes produced from experimental data. ORA-based methods merely count the number of differentially expressed genes in specific functional category gene sets such as the Gene Ontology (GO) (*Blake et al., 2013*), the Kyoto Encyclopedia of Genes and Genomes (KEGG) (*Kanehisa et al., 2016*), BioCarta (*Nishimura, 2001*), and Reactome (*Joshitope et al., 2005*). Then they determine significance of the overlaps via statistical tests such as Fisher's exact test. Many tools are based on this method including Onto-Express (*Draghici et al., 2003*; *Khatri et al., 2002*), Fisher (*Khatri, Sirota & Butte, 2012*), and the Gene Ontology Enrichment Analysis Software Toolkit (GOEAST) (*Zheng & Wang, 2008*). However, ORA-based methods only take into account large changes in individual genes that significantly affect pathways and they do not account for smaller changes in sets of functionally-related genes (i.e., pathways) capable of significant effects. Functional class scoring (FCS) based methods have been used to avoid this limitation of ORA-based methods. FCS-based methods take into account the coordinated gene expression changes in pathways, such as gene set enrichment analysis (GSEA) (*Subramanian et al., 2005*), gene set analysis (GSA) (*Efron & Tibshirani, 2006*), and mean-rank gene set enrichment tests (MRGSE) (*Liu et al., 2008*). However, ORA-based and FCS-based methods are both limited because they do not take into account the complex interactions between genes or the complex topology of pathways. To overcome this limitation, pathway-topology-based methods were proposed. Pathway-topology-based methods integrate the complex interactions between genes using pathway topology information, specifically KEGG signaling pathway information.

Signaling pathway impact analysis (SPIA), one of the most widely-used pathway-topology-based methods, considers both the number of differentially expressed genes (DEGs) in a given pathway and the topology information of that pathway (*Tarca et al., 2009*). Many improved methods based on SPIA have been proposed. *Li et al. (2015)* developed a method called sub-SPIA, which used a minimum spanning tree way to prune signaling pathways and used the SPIA method to identify significant signaling subpathways (*Li et al., 2015*). *Bao et al. (2016)* developed two SPIA-based methods called PSPIA and MSPIA. These two methods replaced +1 or −1 interaction strength in SPIA with the interaction strength of the Pearson correlation coefficients and mutual information, respectively (*Bao et al., 2016*). There are different pathway-topology methods that make use of the topological information of signaling pathways. For instance, Gene Graph Enrichment Analysis (GGEA) uses prior knowledge derived from directed gene regulatory networks (*Geistlinger et al., 2011*). *Liu, Xu & Bao (2019)* used a subgraph method to take advantage of pathway topological information (*Liu, Xu & Bao, 2019*). ROntoTools introduced a term of perturbation factor by considering the type of interactions to take the pathway topology into consideration (*Tarca et al., 2009*;

*Voichita, Donato & Draghici, 2012*). *Sebastian-Leon et al. (2014)* developed a method using topology to detect liner subpathways in a signaling pathway (*Sebastian-Leon et al., 2014*).

These methods still have their disadvantages. Pathway-topology-based methods do not consider the importance of genes in pathways. Gene-weight-based methods have been proposed to overcome this limitation. Pathway analysis with down-weighting of overlapping genes (PADOG) uses the frequency of a present gene in the analyzed pathways to improve gene set analysis (*Tarca et al., 2012*). Functional link enrichment of gene ontology or gene sets (LEGO) measures gene weights in a gene set according to its relative association with genes inside and outside the gene set in a functional association network (*Dong et al., 2016*). *Fang et al. (2017)* proposed an improved SPIA method called SPIA-IS that measured and assigned the importance as the average output degree of the gene in the pathway.

A signaling pathway is a cascade of molecular reactions that bring out the functional attributes (e.g., cell proliferation, apoptosis) associated with the biological behaviors of disease cells using effector genes. Effector genes receive signals without outputting signals to other genes in an individual signaling pathway (*Sebastian-Leon et al., 2014*). Diseases are always related to the abnormal signal that the effector genes receive. Therefore, the signal that the effector genes receive can be very different under disease and normal conditions. The limitation of the previously mentioned methods, including gene-weight-based methods, is that they do not consider the signal variations between disease and normal conditions.

Additionally, the functional attributes in the same signaling pathway may be very different from one another, and can sometimes be opposites. For example, there are two opposite functional attributes on the axon guidance pathway: axon repulsion and axon attraction (see the hsa04360 pathway in the KEGG dataset). We cannot determine which functional attributes contribute more to the disease using most current pathway analysis methods. Furthermore, some pathways consist of several parts, each with very different contributions. For example, the Wnt signaling pathway is significant across different diseases and can be divided into three parts. Most existing pathway analysis methods cannot determine which part of the Wnt signaling pathway most significantly contributes to a specific disease.

We propose a new method that considers the signal variations between normal and disease conditions that effector genes received in pathways: the signaling pathway functional attributes analysis (SPFA) method. SPFA calculates the gene expression changes in a given pathway using an ORA method and then combines the ORA method results with the signal variation results under two conditions (normal vs. disease). The signal variations can help identify functional attributes and abnormal pathways. We tested the capabilities of the proposed signaling pathway analysis method on a series of real datasets using three parameters. We also showed that the two types of probabilities considered in this method were indeed independent. Ultimately, we verified the usefulness of the signal variations the effector genes received under two different conditions using the SPFA method.

## MATERIALS AND METHODS

### Data sources and preprocessing

Signaling pathway analysis methods require two types of input: a collection of pathways and a list of genes or gene products with accompanying expression values across different samples between the compared phenotypes. We used the KEGG signaling pathway as it is the most common manually-curated signaling pathway used for pathway analysis. We downloaded 213 signaling pathways from the KEGG PATHWAY dataset.

We acquired 33 disease gene expression datasets from the KEGGdzPathwaysGEO R-package and KEGGandMetacoreDzPathwaysGEO R-packages (Table 1) (*Tarca, Bhatti & Romero, 2013*; *Tarca et al., 2012*). Each disease gene expression dataset was matched with a corresponding disease KEGG pathway. For example, a colorectal cancer dataset was associated with the colorectal cancer pathway (*Tarca et al., 2012*). The corresponding disease KEGG pathways were called target pathways. Three rules were used to select the gene expression datasets:

1. The dataset's DEGs were available. If no DEGs were selected, other comparable methods would return null results.

2. The results of these datasets could be analyzed. Pathway analysis result *p*-values of 1 could not be analyzed.

3. The target pathways of these datasets were KEGG pathways since we used KEGG pathways as examples.

DEGs were selected if they contained more than 200 genes with FDR adjusted *p*-values < 0.05. Otherwise, we selected more than 200 genes with original *p*-values < 0.05 and absolute log (fold change) > 1.5. If DEGs still less than 200 genes, we selected the top 1% of genes ranked by *p*-values as DEGs.

### SPFA algorithm design

To assess the signal variations between two conditions (normal vs. disease) that the effector genes received from upstream genes, we calculated the sum of signal variations from all upstream genes to effector genes. Given an effector gene $g_e$ and an upstream gene $g_s$, the signal variation from the gene $g_s$ to the effector gene $g_e$ can be defined as:

$$e_{se} = \frac{\left| cor^{\text{disease}}(g_s g_e) - cor^{\text{normal}}(g_s g_e) \right|}{d_{se}} \tag{1}$$

where $cor^{\text{disease}}(g_s g_e)$ and $cor^{\text{normal}}(g_s g_e)$ refer to the Pearson correlation coefficient between the gene expression data of gene $g_s$ and gene $g_e$ in the disease and normal states, respectively. $d_{se}$ is the network distance between gene $g_s$ and gene $g_e$. The Pearson correlation coefficient is always used in gene co-expression networks to represent the strength of interactions between two genes. The Pearson correlation coefficient can also be used to represent the strength of an interaction between two gene expression values. Studies have shown that the genetic regulatory patterns in signaling pathways between
**Table 1  Data sets used for assessing the proposed method and compared methods.**

| ID | Target pathway | GEO ID | References |
|----|----------------|--------|------------|
| 1 | Colorectal cancer | GSE4107 | *Hong et al. (2007)* |
| 2 | Colorectal cancer | GSE4183 | *Galamb et al. (2008)* and *Gyorffy et al. (2009)* |
| 3 | Colorectal cancer | GSE8671 | *Sabates-Bellver et al. (2007)* |
| 4 | Colorectal cancer | GSE9348 | *Hong et al. (2010)* |
| 5 | Colorectal cancer | GSE23878 | *Uddin et al. (2011)* |
| 6 | Non-small cell lung cancer | GSE18842 | *Sanchez-Palencia et al. (2010)* |
| 7 | Pancreatic cancer | GSE15471 | *Badea et al. (2008)* |
| 8 | Pancreatic cancer | GSE16515 | *Pei et al. (2009)* |
| 9 | Pancreatic cancer | GSE32676 | *Donahue et al. (2012)* |
| 10 | Thyroid cancer | GSE3467 | *He et al. (2005)* |
| 11 | Thyroid cancer | GSE3678 | – |
| 12 | Alzheimer's disease | GSE5281_HIP | *Liang et al. (2007)* |
| 13 | Alzheimer's disease | GSE5281_EC | *Liang et al. (2007)* |
| 14 | Alzheimer's disease | GSE5281_VCX | *Liang et al. (2007)* |
| 15 | Alzheimer's disease | GSE1297 | *Blalock et al. (2004)* |
| 16 | Alzheimer's disease | GSE16759 | *Juan et al. (2010)* |
| 17 | Chronic myeloid leukemia | GSE24739_G0 | *Affer et al. (2011)* |
| 18 | Chronic myeloid leukemia | GSE24739_G1 | *Affer et al. (2011)* |
| 19 | Acute myeloid leukemia | GSE14924_CD4 | *Le Dieu et al. (2009)* |
| 20 | Acute myeloid leukemia | GSE14924_CD8 | *Le Dieu et al. (2009)* |
| 21 | Acute myeloid leukemia | GSE9476 | *Stirewalt et al. (2008)* |
| 22 | Dilated cardiomyopathy | GSE1145 | – |
| 23 | Dilated cardiomyopathy | GSE3585 | *Barth et al. (2006)* |
| 24 | Endometrial cancer | GSE7305 | *Hever et al. (2007)* |
| 25 | Glioma | GSE19728 | *Liu et al. (2011)* |
| 26 | Glioma | GSE21354 | *Liu et al. (2011)* |
| 27 | Huntington's disease | GSE8762 | *Runne et al. (2007)* |
| 28 | Parkinson's disease | GSE20291 | *Zhang et al. (2005)* |
| 29 | Parkinson's disease | GSE20164 | *Zheng et al. (2010)* |
| 30 | Prostate cancer | GSE6956AA | *Wallace et al. (2008)* |
| 31 | Prostate cancer | GSE6956C | *Wallace et al. (2008)* |
| 32 | Renal cell carcinoma | GSE781 | *Lenburg et al. (2003)* |
| 33 | Renal cell carcinoma | GSE14762 | *Wang et al. (2009)* |

genes are different under normal and disease conditions (*Jung, 2018*). If the genetic regulatory pattern between the two genes changes, the signal transmitted between the two genes will be very different. Thus, we used the Pearson correlation coefficient to calculate the signal variations that the effector genes received from their upstream genes. However, if an upstream gene does not directly transmit a signal, the signal may be attenuated. Therefore, we used the network distance $d_{se}$ between gene $g_s$ and gene $g_e$ as a penalty coefficient.

For each effector gene $g_i$ in a given pathway, the accumulated signal variations between normal and disease conditions that the upstream genes received (total $s$ genes in the upstream of the gene $g_i$) were calculated using the formula:

$$ASV(g_i) = \sum_{j=1}^{s} e_{ij} \tag{2}$$

The accumulated signal variation $ASV(g_i)$ of the effector gene $g_i$ in a pathway can help us distinguish among the functional disease attributes. Effector genes with high $ASV(g_i)$ demonstrate that these functional attributes significantly contribute to their corresponding diseases.

For a given signaling pathway, the total accumulated signal variation ASV can be defined as:

$$ASV = \sum_{i=1}^{k} ASV(g_i) \tag{3}$$

where $k$ is the total number of effector genes in the given pathway.

Ultimately, the probability $P_{sd}$ used to measure the signal variations between two conditions (normal vs. disease) that those effector genes received from genes upstream in a given signaling pathway $P_x$ is based on the pathway's $ASV(P_x)$. The same number of genes as the one observed on the given signaling pathway are randomly selected from all genes (random gene IDs) and have any possible expression data in all samples in the range of the experimenter. Therefore, the observed signal variations were obtained by permuting the gene IDs 2000 times. $ASV_{per}(P_x)$ was the total accumulated signal variation of the given pathway $P_x$ obtained in the $per$-th time. The probability $P_{sd}(P_x)$ of the given pathway was calculated as:

$$P_{sd}(P_x) = \frac{\sum I(ASV_{per}(P_x) \geq ASV(P_x))}{2000} \tag{4}$$

where $I$ is a function that returns 1 when the argument is true and 0 when the argument is false.

The probability $P_{sd}$ does not measure the gene differential expression in a given pathway. Thus, we had to combine the probability $P_{sd}$ with the probability $P_{de}$ which can measure the total gene differential expression in a given signaling pathway. The probability $P_{de}$ of a given pathway $P_x$ can be calculated through the following hypergeometric test:

$$P_{de}(P_x) = 1 - \frac{\binom{t}{r}\binom{m-t}{n-r}}{\binom{m}{n}} \tag{5}$$

where the whole genome contains a total of $m$ genes, $n$ genes are the number of DEGs in the $m$ genes, and the given pathway contains $t$ genes and $r$ DEGs.

The probability $P_{sd}$ uses the Pearson correlation coefficient of the two genes' expression data, but the probability $P_{de}$ uses the number of DEGs in a pathway. Thus, the two probabilities are independent of each other. The significance of the given pathway was calculated following the SPIA method which combines the probabilities $P_{sd}$ and $P_{de}$ (*Tarca et al., 2009*). The formulas are:

$$P = c - c \cdot \ln(c) \tag{6}$$

$$c = P_{sd} \times P_{de} \tag{7}$$

where $c$ is a product of $P_{de}$ and $P_{sd}$. $P$ is the combined probability of the signaling pathway.

## Significantly enriched pathway analysis using SPFA

The SPFA procedure identifies significantly enriched pathways in two steps (Fig. 1). The first step measures the total gene differential expression in the signaling pathways. DEGs need to first be identified from the gene expression datasets. Then the DEGs are mapped onto the signaling pathways. Finally, the signaling pathway $p$-values are calculated using a hypergeometric test.

The second step is to measure the signal variations between the two conditions (normal vs. disease) that effector genes received from upstream genes in the signaling pathways. This is completed by:

1. Finding all effector genes in each signaling pathway.
2. Ascertaining all paths from the upstream genes to the effector genes in each signaling pathway. If a path exists between the upstream genes and effector genes, an interaction must exist between them. The path's network distances are used to weight the corresponding interactions.
3. Using the Pearson correlation coefficient absolute difference values between the disease and normal samples to calculate the signal variations of the corresponding interactions.
4. Using the network distance of each interaction to decrease their signal variations.
5. Calculating the accumulation of the signal variations between the effector genes and upstream genes for each effector gene.
6. Calculating the sum of the accumulations of all effector genes in each signaling pathway.
7. Evaluating the statistical significance of each pathway based on their score.

Ultimately, the results of the two steps are combined into one $p$-value. We used the FDR adjust method on the combined $p$-value to determine the significance of each signaling pathway. The pathways with the adjusted combined $p$-values smaller than a threshold value were considered as significant pathways.

## The distribution of effector genes in the signaling pathways

Studying the signal variations between two conditions (normal vs. disease) that the effector genes received leads to a deeper understanding of the biological behaviors of disease cells. Effector genes are widely scattered throughout the signaling pathways. If a gene has no

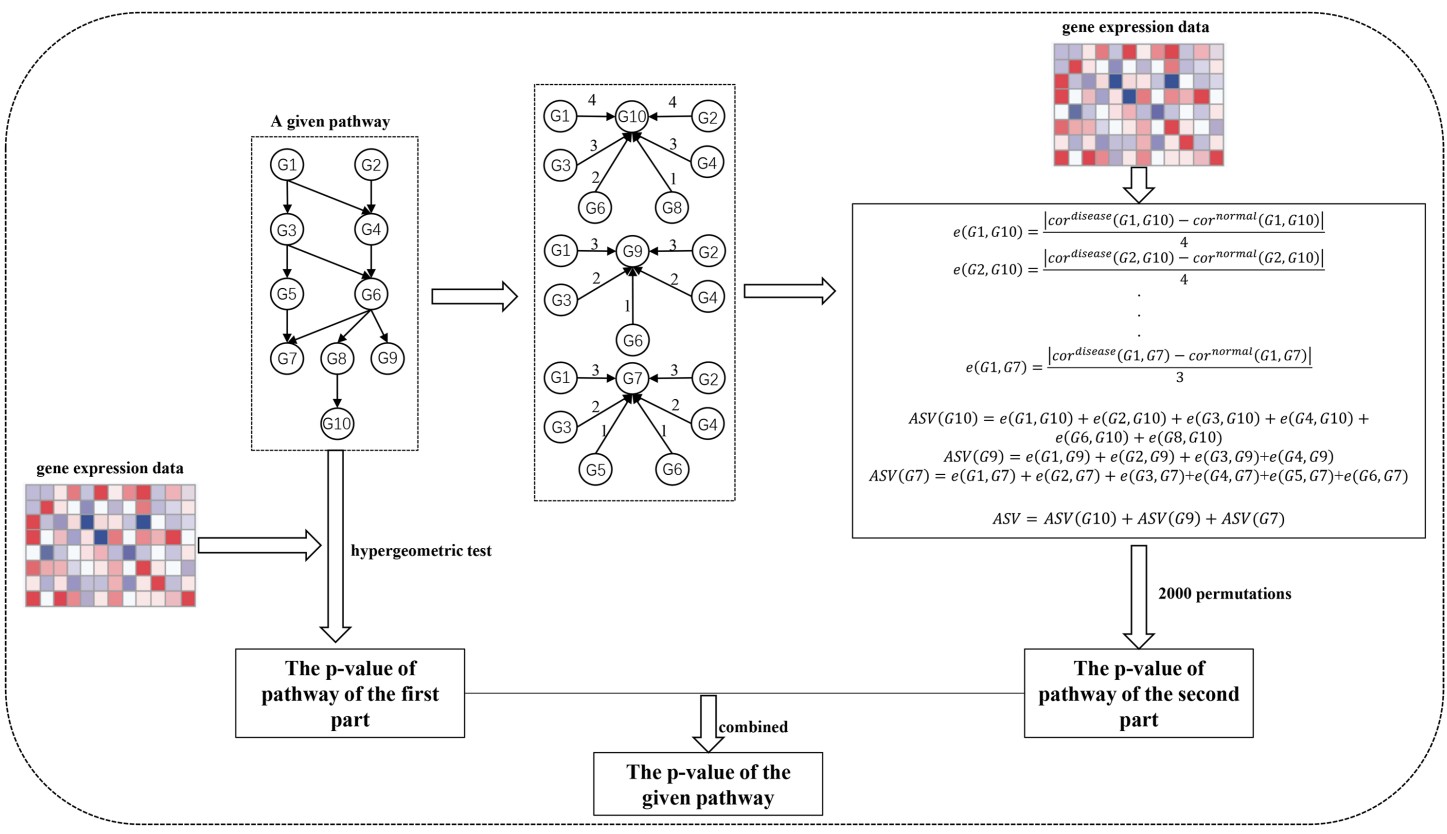

**Figure 1** **The workflow of SPFA method.** The step by step to identify significant signaling pathways using SPFA.

signal inputs in an individual signaling pathway, the gene is not considered an effector gene. The distribution of effector genes in each signaling pathway can be seen in Fig. 2. One hundred and ninety-five of the 213 signaling pathways contained effector genes.

## Comparison methods and measures

We compared seven methods to SPFA, including Fisher (*Khatri, Sirota & Butte, 2012*), GSA (*Efron & Tibshirani, 2006*), GSEA (*Subramanian et al., 2005*), MRGSE (*Liu et al., 2008*), SPIA (*Ullah, 2013*), ROnoTools (*Tarca et al., 2009*; *Voichita, Donato & Draghici, 2012*), and PADOG (*Tarca et al., 2012*). We selected these methods for their stability and prevalence; they can be compared using the same R environment as SPFA.

There is no universally accepted technique for the validation of the results of pathway analysis methods. Current pathway analysis methods use the results of a very small number of datasets based on searching corresponding published life literature. This approach has its limitations. First, the number of datasets used is small. Second, authors often search their own, leading to biased results. Third, complex biological phenomena always directly or indirectly correspond to multiple signaling pathways.

*Tarca et al. (2012)* compiled an objective and reproducible approach based on multiple datasets (*Tarca et al., 2012*). This approach avoided a biased literature search and required testing on a large number of different datasets (at least 10). In this work, we followed

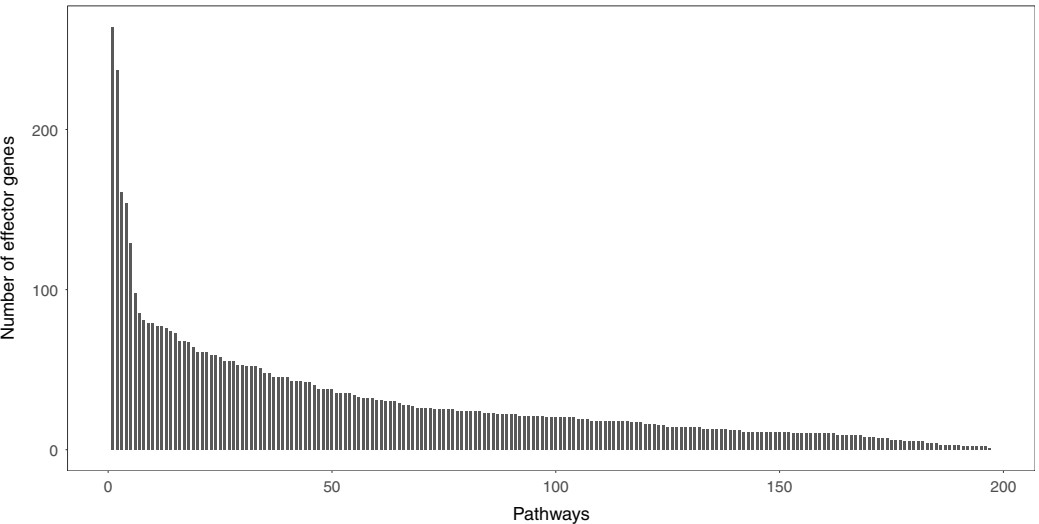

**Figure 2 The distribution of the effector genes' number in each signaling pathway.** A total of 195 of 213 signaling pathways contain the effector genes.

*Tarca et al. (2012)* validation approach. Two measurements were compared in this validation approach. The first measurement was the median *p*-value of the 33 target pathways of the 33 disease datasets. Smaller median *p*-values meant higher sensitivity. The second measurement was the median rank of the 33 disease target pathways. The higher ranked methods were more accurate. To further validate the different pathway analysis method results, we used a third measurement: the ratio of significant pathways (using a significance threshold of 0.05 of the adjusted *p*-value) in the 33 datasets. This measured the method's ability to control false positive and false negative rates.

# RESULTS

## The independence between the two probabilities

The two probabilities $P_{de}$ and $P_{sd}$ are theoretically independent under the null hypothesis. We verified their independence by calculating the squared correlation coefficient between the two probabilities using the 33 gene expression datasets (Table 2). Our results showed that the average squared correlation coefficient of the 33 datasets was $R^2 = 0.029$. Only four of the 33 squared correlation coefficients were slightly higher than $R^2 = 0.09$. These results indicated essentially no correlation between the two probabilities.

## SPFA method performance

We compared SPFA with the other seven methods using three measurements: the median *p*-value of the 33 target pathways, the median rank of the 33 target pathways, and the ratio of significant pathways. The signaling pathways with adjusted *p*-values ≤ 0.05 were significant.

When comparing the median rank of the 33 target pathways, SPFA ranked first (Fig. 3). When comparing the median *p*-value of the 33 target pathways, SPFA ranked fourth

**Table 2 The squared correlation coefficients between the two probabilities using the 33 gene expression datasets.** The four squared correlation coefficients which are slightly more than 0.09 are shown in bold.

| GEO ID | Squared correlation between the probabilities $P_{de}$ and $P_{sd}$ |
|---|---|
| GSE4107 | 0.006928102 |
| GSE4183 | 0.032207913 |
| GSE8671 | 0.00011503 |
| GSE9348 | 0.027441819 |
| GSE23878 | 0.013047606 |
| GSE18842 | 0.089945631 |
| GSE15471 | 0.032082501 |
| GSE16515 | 0.022817456 |
| GSE32676 | 0.010161372 |
| GSE3467 | 0.001098836 |
| GSE3678 | 0.000879454 |
| GSE5281_HIP | 0.026379598 |
| GSE5281_EC | 0.032472155 |
| GSE5281_VCX | 0.063438794 |
| GSE1297 | 0.000346566 |
| GSE16759 | 0.028461474 |
| GSE24739_G0 | 0.009721816 |
| GSE24739_G1 | 0.022257943 |
| GSE14924_CD4 | **0.106127** |
| GSE14924_CD8 | 0.051189135 |
| GSE9476 | 0.073960111 |
| GSE1145 | **0.098132151** |
| GSE3585 | 6.61523E−05 |
| GSE7305 | **0.101902794** |
| GSE19728 | **0.094956883** |
| GSE21354 | 0.00854786 |
| GSE8762 | 0.000830428 |
| GSE20291 | 0.000499751 |
| GSE20164 | 7.48134E—07 |
| GSE6956AA | 0.006999771 |
| GSE6956C | 0.001917359 |
| GSE781 | 0.000219909 |
| GSE14762 | 0.000513602 |
| Average | 0.029262658 |

(Fig. 4). Notably, the methods with the highest ranking in one measurement did not necessarily rank the highest in the others. This is because different measurements analyze different abilities. For example, MRGSE was first in median $p$-value but was sixth in median rank. Fisher was second in median $p$-value but ranked fourth in median rank. To better compare SPFA's performance against the other methods, we added the ranks of

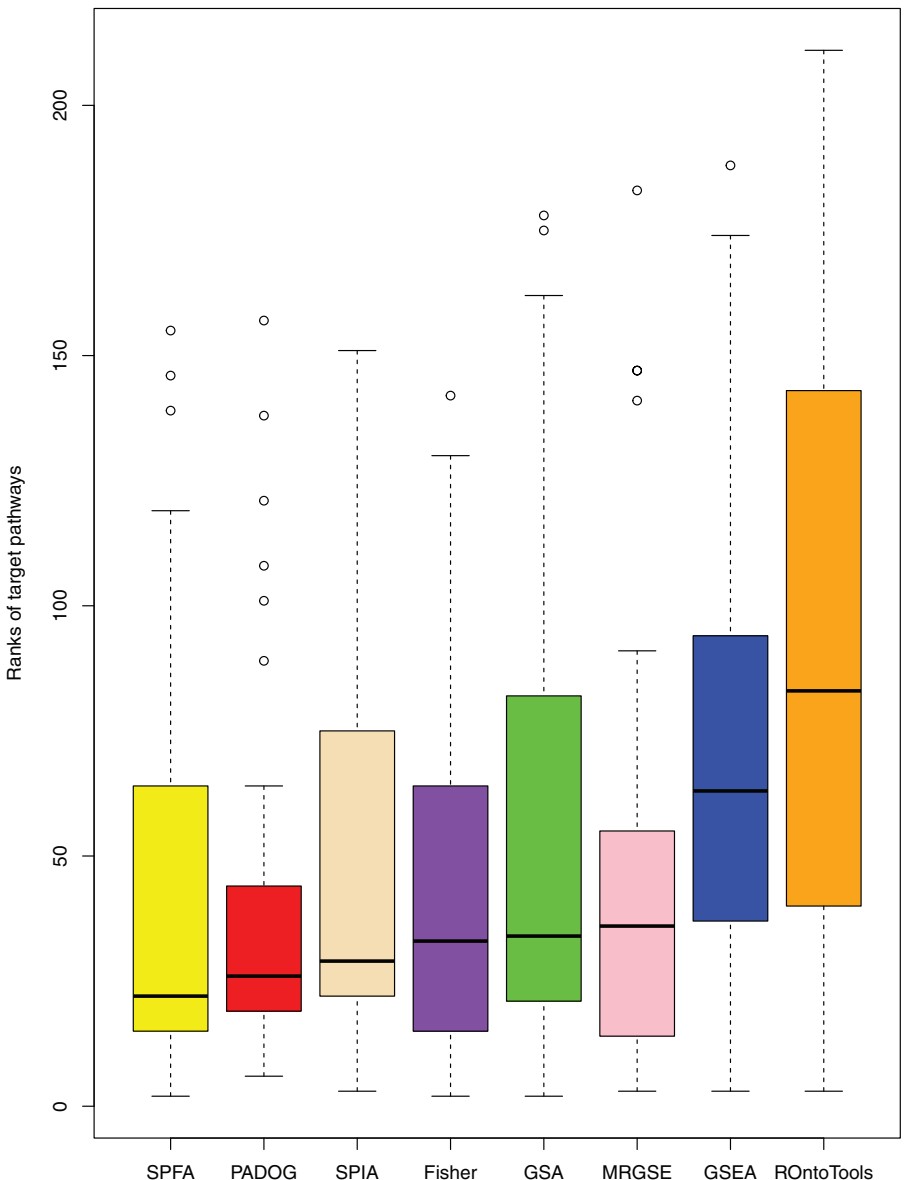

**Figure 3 The distribution of the target pathways ranks of the eight methods using 33 datasets.** SPFA performs the 1st among eight methods in terms of the median ranks of the 33 target pathways.

the median $p$-value and median rank values from each method together. We found that the combined value of SPFA and PADOG was the smallest (Table 3).

To further assess the performance of the eight methods, we collected the results from other general pathways typically associated with cancer using the 18 out of 33 datasets with a form of cancer in Table 4: Apoptosis and Pathways in cancer. When using the Apoptosis pathway and Pathway in cancer pathway instead of target pathways, SPFA's median ranks were both first, and the median $p$-values of MRGSE were also both ranked first. These results were in alignment with the target pathway results. However, when using

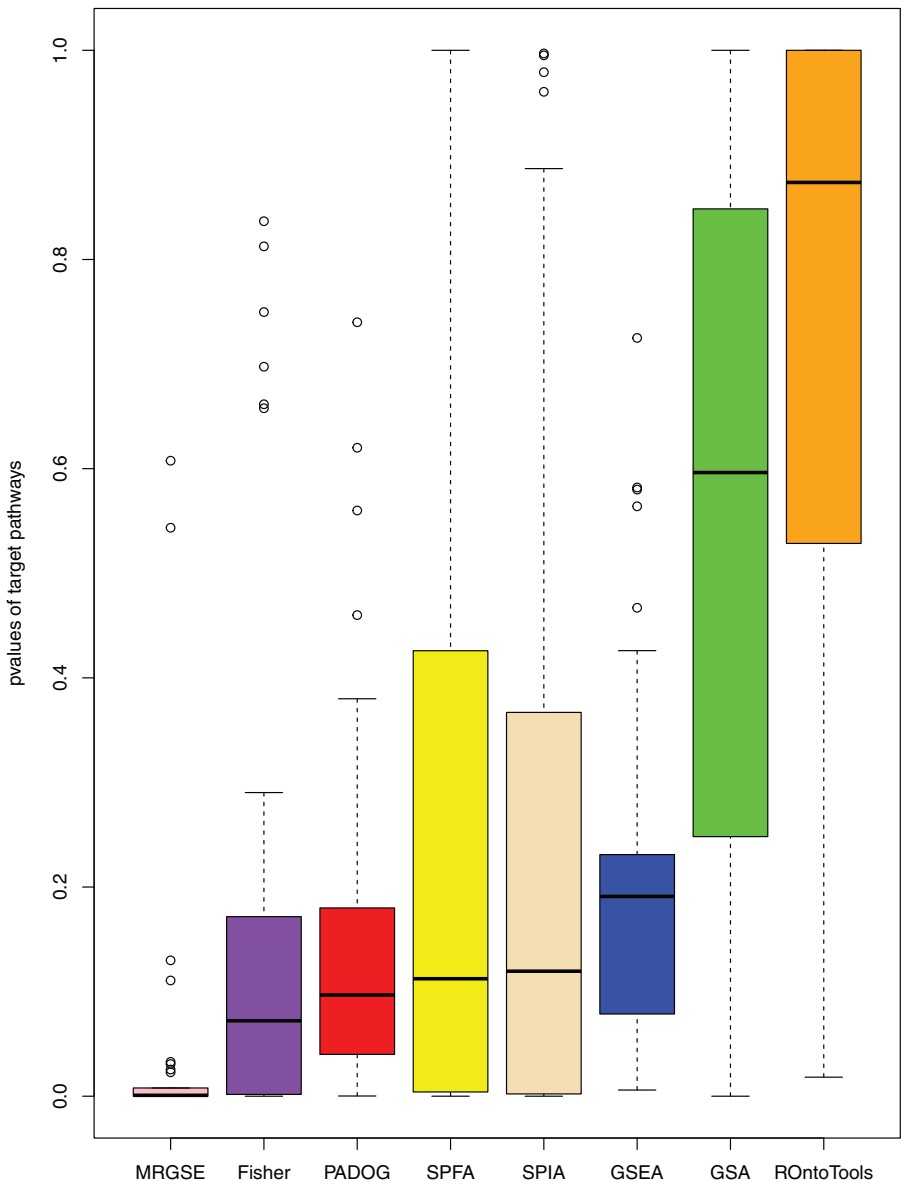

**Figure 4 The distribution of the target pathways _p_-values of the eight methods using 33 datasets.**
SPFA performs the 4th among eight methods in terms of the median _p_-values of detecting the 33 tar-
get pathways.

the Apoptosis pathway and Pathway in cancer pathway instead of the target pathways,
PADOG's median _p_-values were both ranked fifth. When using the Apoptosis pathway,
SPFA's median _p_-value ranked third. When using the Pathway in cancer pathway, SPFA's
median _p_-value ranked fourth. All these results suggest that SPFA had the best accuracy
and a good sensitivity when compared with the other seven methods.

Additionally, our results showed that SPFA's ratio of significant pathways was
moderate, 0.16 (Fig. 5), compared to the others. MRGSE's ratio of significant pathways was
almost 0.5, and it could be questioned whether a such number of pathways was realistic.

Table 3 **The combined rank values of the ranks in terms of the median *p*-values and the median ranks of target pathways of eight methods.**

|  | Methods | Ranks of the median *p*-values | Ranks of the median ranks | Sum |
|---|---|---|---|---|
| 1 | SPFA | 4 | 1 | 5 |
| 2 | PADOG | 3 | 2 | 5 |
| 3 | Fisher | 2 | 4 | 6 |
| 4 | MRGSE | 1 | 6 | 7 |
| 5 | SPIA | 5 | 3 | 8 |
| 6 | GSA | 7 | 5 | 12 |
| 7 | GSEA | 6 | 7 | 13 |
| 8 | ROnoTools | 8 | 8 | 16 |

Table 4 **The results of other general pathways: apoptosis and pathway in cancer typically associated with cancer using the 18 out of 33 datasets with a form of cancer.** For each pathway, the values for the type of methods with the smallest median *p*-values and ranks (strongest association with the phenotype) are shown in bold.

| Pathway statistic | Apoptosis | | Pathway in cancer | |
|---|---|---|---|---|
| | *p*-Values median | Ranks median | *p*-Values median | Ranks median |
| SPFA | 0.0658 | **39.5** | 7.94E−05 | **3** |
| Fisher | 0.0235 | 46 | 2.25E−05 | 4 |
| SPIA | 0.0661 | 53 | 1.62E−05 | 5 |
| GSA | 0.779 | 125 | 0.539 | 44.5 |
| GSEA | 0.393 | 116.5 | 0.291 | 102 |
| MRGSE | **0.00213** | 46 | **2.7E−08** | 3 |
| RontoTools | 0.647 | 70.5 | 1 | 210 |
| PADOG | 0.26 | 71 | 0.09 | 24 |

GSA's ratio of significant pathways was lower than 0.05, and it reflected that the GSA method had a high false negative rate. The methods had a modest ratio of significant pathways indicated that the method had a modest false positive rate and a modest false negative rate. Thus, the discriminative ability of SPFA was good when compared with the other seven methods. In conclusion, our results strongly supported that SPFA was well-suited for signaling pathway analysis and confirmed previously reported results in *Dong et al. (2016)*.

## Sources of improvement for SPFA

The main source of improvement in SPFA is that it uses signal variations that effector genes received under normal and disease conditions. SPFA is compared to the simpler ORA-based method used to calculate the probability $P_{de}$ without accounting for signal variations (Fig. 6). As shown in Fig. 6, the ORA-based method has a higher (worse) rank and *p*-value than SPFA for the target pathways.

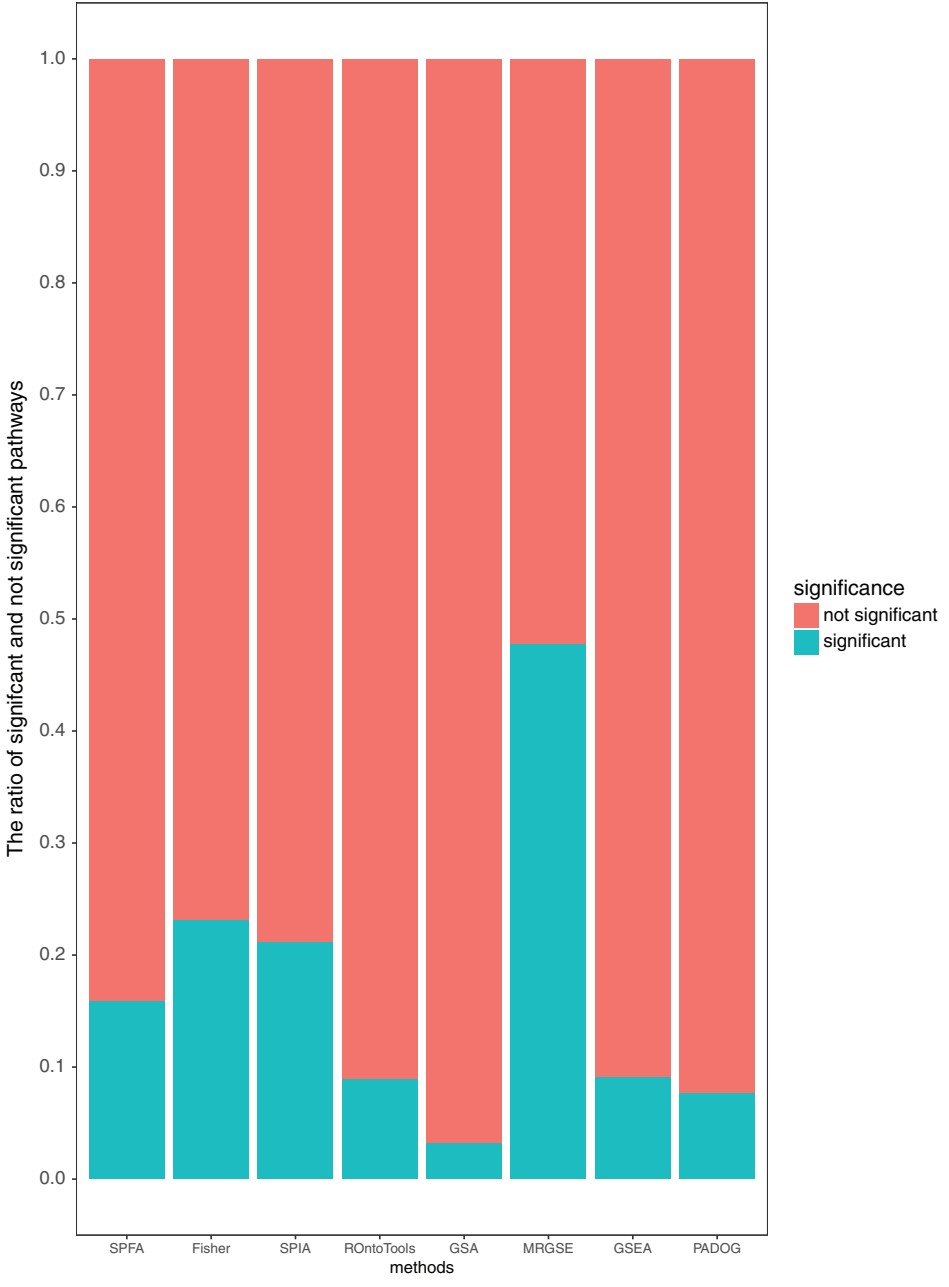

**Figure 5 Average percentage of the pathways detected as significant and not significant by each method using the threshold of *p*-values ≤ 0.05.**

## Validating the correlation between diseases and the signal variations that effector genes received under two different conditions

To validate the correlation between diseases and the signal variations that effector genes received under two different conditions (normal vs. disease), we analyzed a colorectal cancer dataset (GSE4183) and an Alzheimer's disease dataset (GSE16759). The colorectal cancer microarray GSE4183 (Affymetrix array HG-U133 Plus2.0) included 15 colorectal cancer samples and 8 normal samples (*Galamb et al., 2008*; *Gyorffy et al., 2009*).

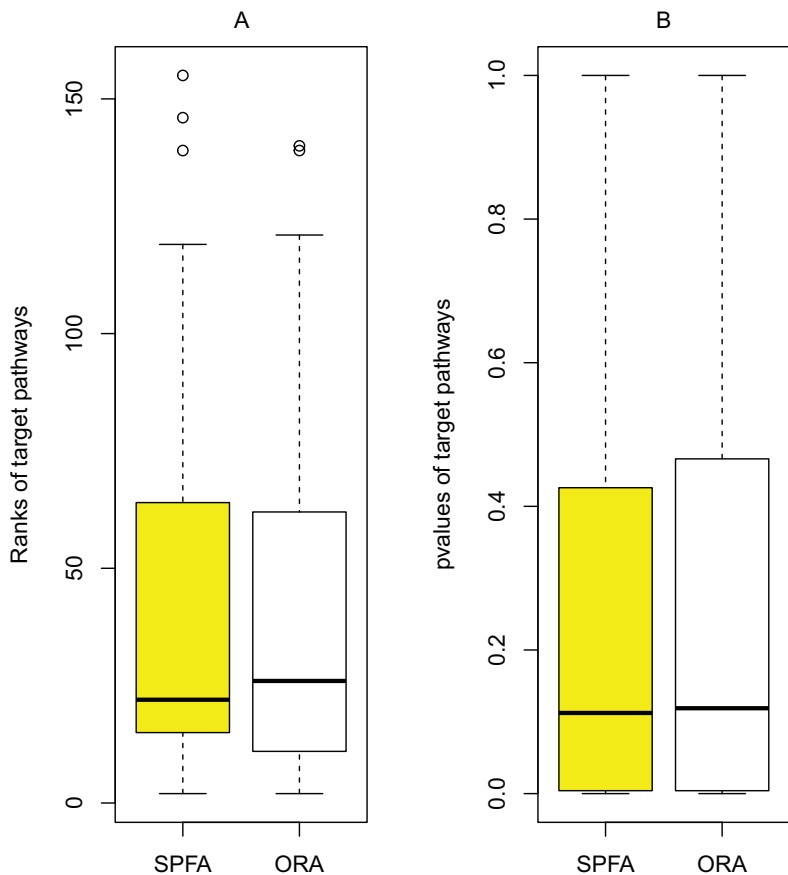

**Figure 6** Determining the contribution of signal variations received by effector genes between two different conditions (normal vs. disease) in SPFA performance. The boxplots show the distribution of the target pathways ranks (A) and *p*-values (B).               

The Alzheimer's disease dataset GSE16759 included four disease samples and four normal samples (*Juan et al., 2010*).

The Wnt signaling pathway was altered in 90% of the colorectal cancer samples (*Galamb et al., 2008*). We assessed the signal variations that effector genes received in the Wnt signaling pathway using the GSE4183 dataset (Fig. 7). The results of (*Galamb et al., 2008*) coincided with our signal variation results (*Galamb et al., 2008*) reported that overexpression of TNS1 could induce the activation of JNK (ENTREZID: 5599, 5601, and 5602). The signal variation that the effector gene ENTREZID: 5602 received ranked first in our results. *Galamb et al. (2008)* detected that RBMS1 is another overexpressed gene and modulator of c-myc (ENTREZID: 4609). c-myc can regulate cell cycles and cause cells to transform pathways. The signal variation that the effector gene ENTREZID: 4609 received ranked second in our results. *Galamb et al. (2008)* also identified that TCF4 is an overexpressed gene that can participate in the transcriptional regulation of genes associated with colon carcinogenesis. These colon carcinogenesis associated genes include c-myc (ENTREZID: 4609), cy-clin D1 (ENTREZID: 595), PPARδ (ENTREZID: 5467), and MMP7 (ENTREZID: 4316). The signal variations that these effector genes received ranked second, fourth, fifth, and sixth, respectively.
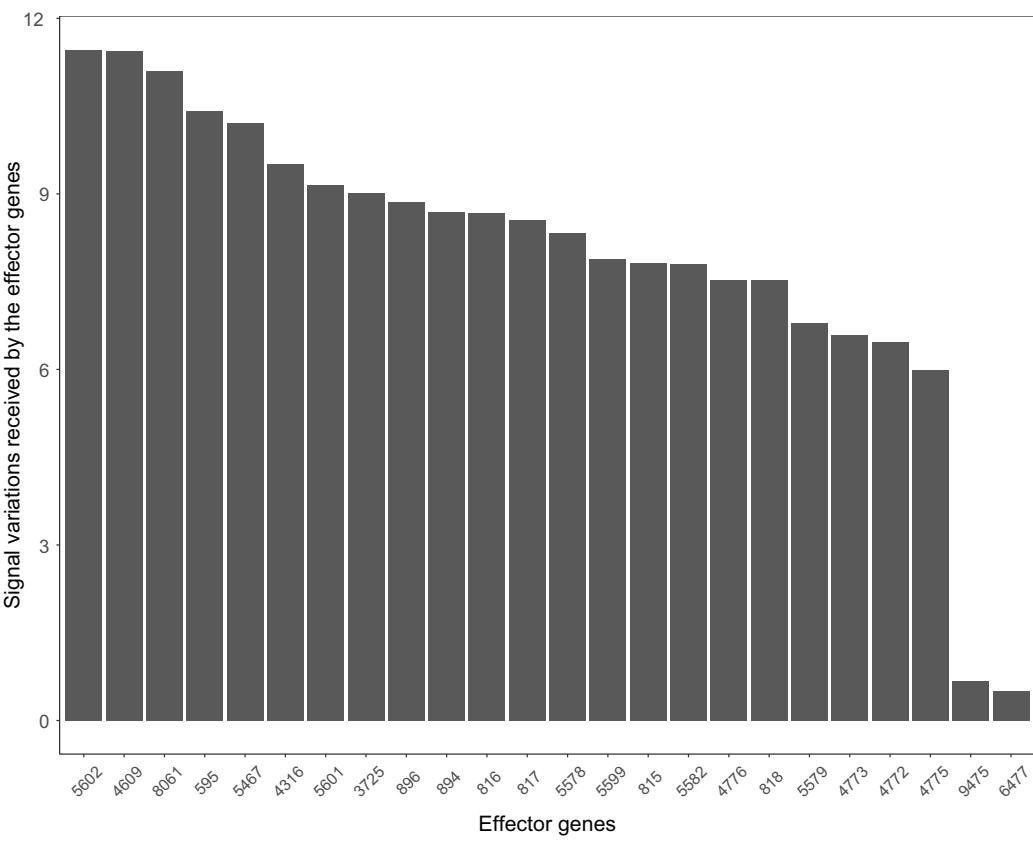

**Figure 7 The signal variations received by effector genes from the upstream genes in the Wnt signaling pathway using colorectal cancer datasets (GSE4183).**

Many pathways can be studied in colorectal cancer datasets. For example, the PI3K-Akt signaling pathway plays a critical role in the growth and progression of colorectal cancer (*Johnson et al., 2010*). The effector genes ENTREZID:596, ENTREZID:842, and ENTREZID:1027 have the highest signal variations and are linked to cell cycle progression and cell survival (Fig. 8). The GSE4183 dataset results further confirmed the role of this pathway in colorectal cancer development.

The Wnt signaling pathway is also closely related to the occurrence and development of Alzheimer's disease (*Inestrosa et al., 2007*). The signal variations that different effector genes received calculating based on the Alzheimer's disease dataset GSE16759 in the Wnt signaling pathway were shown in Fig. 9. The signal variations that the effector genes: ENTREZID: 595 and 896 received were considerably higher than the other effector genes in the Wnt signaling pathway. This result validated evidence of crosstalk between the Alzheimer's disease signaling pathway and the two effector genes' upstream genes in the Wnt signaling pathway.

All these results indicated the high correlation between diseases and the signal variations calculated using the SPFA method.

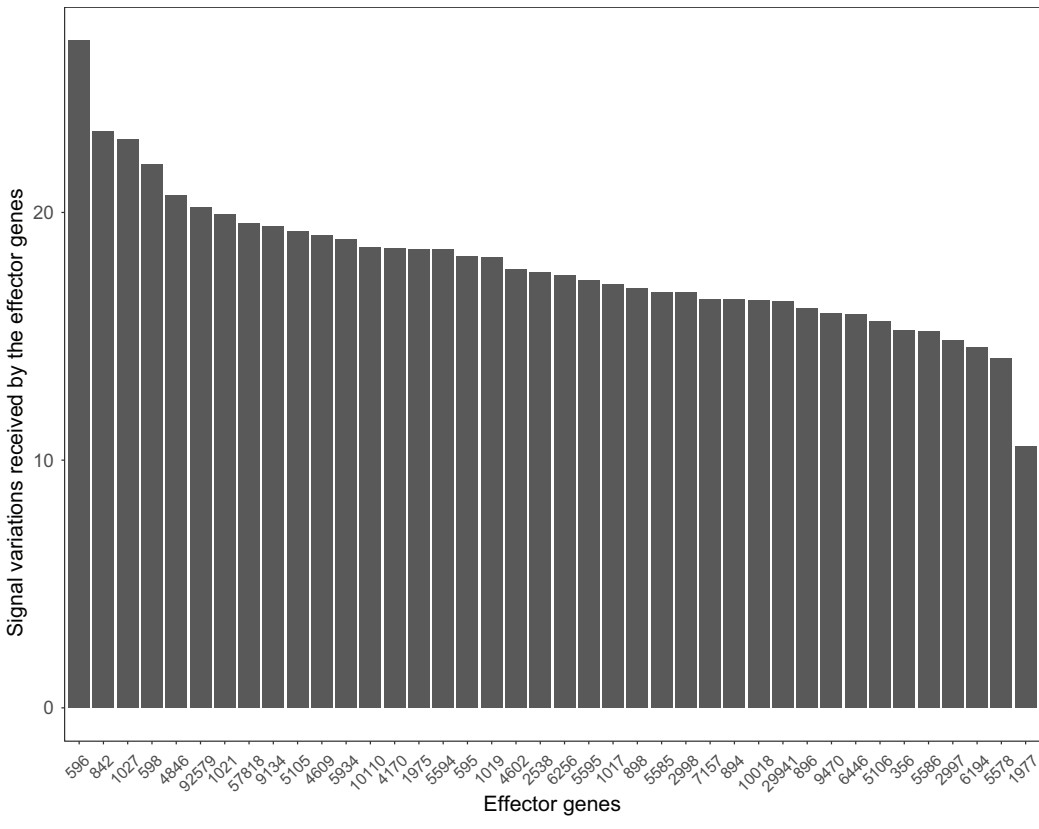

**Figure 8 The signal variations received by effector genes from the upstream genes in the PI3K-Akt signaling pathway using colorectal cancer datasets (GSE4183).**

## The other usages of the signal variations that effector genes received under two different conditions

The signal variations that effector genes received under two different conditions can show the different contributions of different functional attributes contributed to their corresponding diseases. We can also identify which parts of the pathway contribute to their corresponding diseases through the signal variations that effector genes received.

When looking at the Wnt signaling pathway results of GSE4183 (Fig. 7), first, we know the functional attributes participating in the cell cycle have abnormal signal variations because most effector genes with high signal variations participate in the pathway cell cycle (including c-myc (ENTREZID: 4609), cy-clin D1 (ENTREZID: 595, 894, and 896), PPARδ (ENTREZID: 5467), and MMP7 (ENTREZID: 4316)). Second, we can know that the abnormal state of the first and second parts of the Wnt signaling pathway may contribute more to colorectal cancer because that the effector genes with high signal variations are all in the two parts. If we were only to observe DEG distribution in the Wnt signaling pathway using the GSE4183 dataset, we would not know which abnormal part contributed to the disease (Fig. 10). Through the result of the Wnt signaling pathway in GSE16759 (Fig. 9), on one hand, according to this result, we can know that the functional attributes linked with the effector genes: ENTREZID: 595 and 896 which had the highest signal

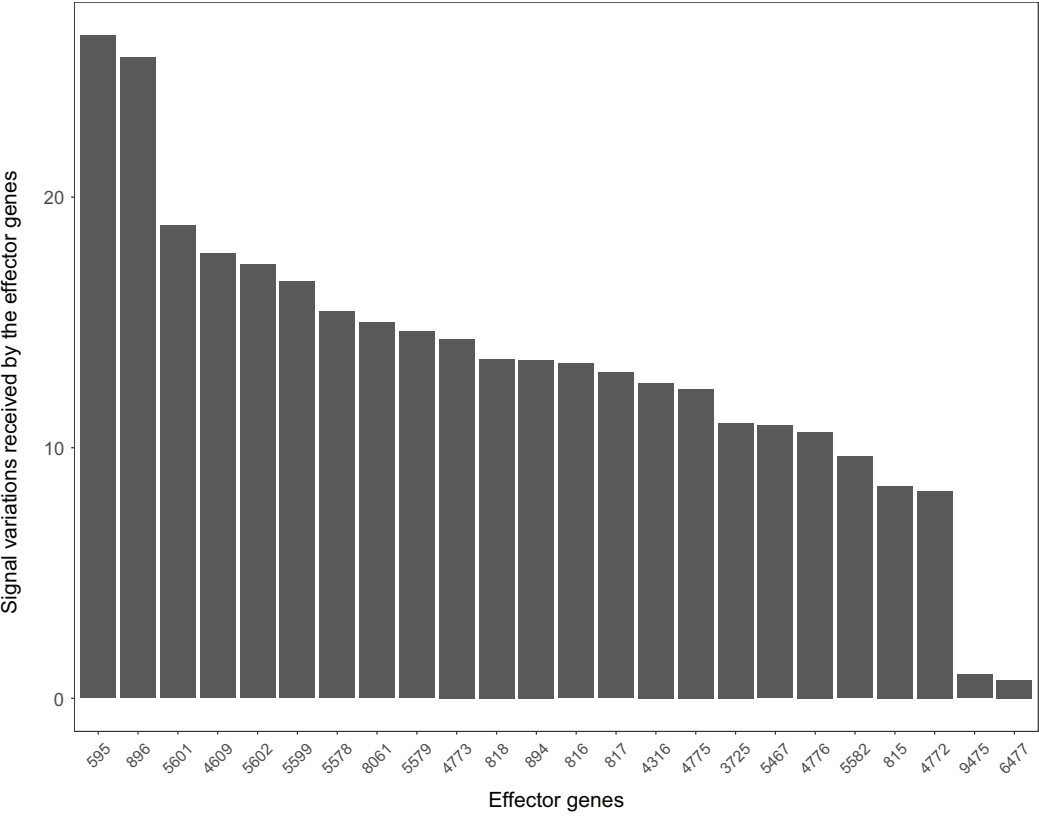

**Figure 9 The signal variations received by effector genes from the upstream genes in the Wnt signaling pathway using Alzheimer's disease datasets (GSE16759).**

variations were abnormal in Alzheimer's disease. On the other hand, this may dominate that the first part of the Wnt signaling pathway may be more related to the occurrence and development of Alzheimer's disease because of crosstalk between the Alzheimer's disease pathway and the first part of the Wnt signaling pathway contained the two effector genes: ENTREZID: 595 and 896.

## DISCUSSION

Functional attributes (associated with biological behaviors of disease cells) are the responses that effector genes respond to the signal they received. Disease cells always have abnormal functional attributes. Thus, the signal that the effector genes received can be very different. However, no current pathway analysis method takes this factor into consideration. Most pathway analysis methods only include the activation and significance of pathways. Their results give us inadequate information on functional attributes that can help explain the biological behaviors of disease cells. Here, we proposed SPFA, a novel signaling pathway analysis method that takes into account signal variations that effector genes receive under disease and normal conditions. Our results showed that SPFA was comparable to seven other signaling pathway analysis methods. We also found that the signal variations that effector genes receive can reflect the contribution of different

# wnt signaling pathway

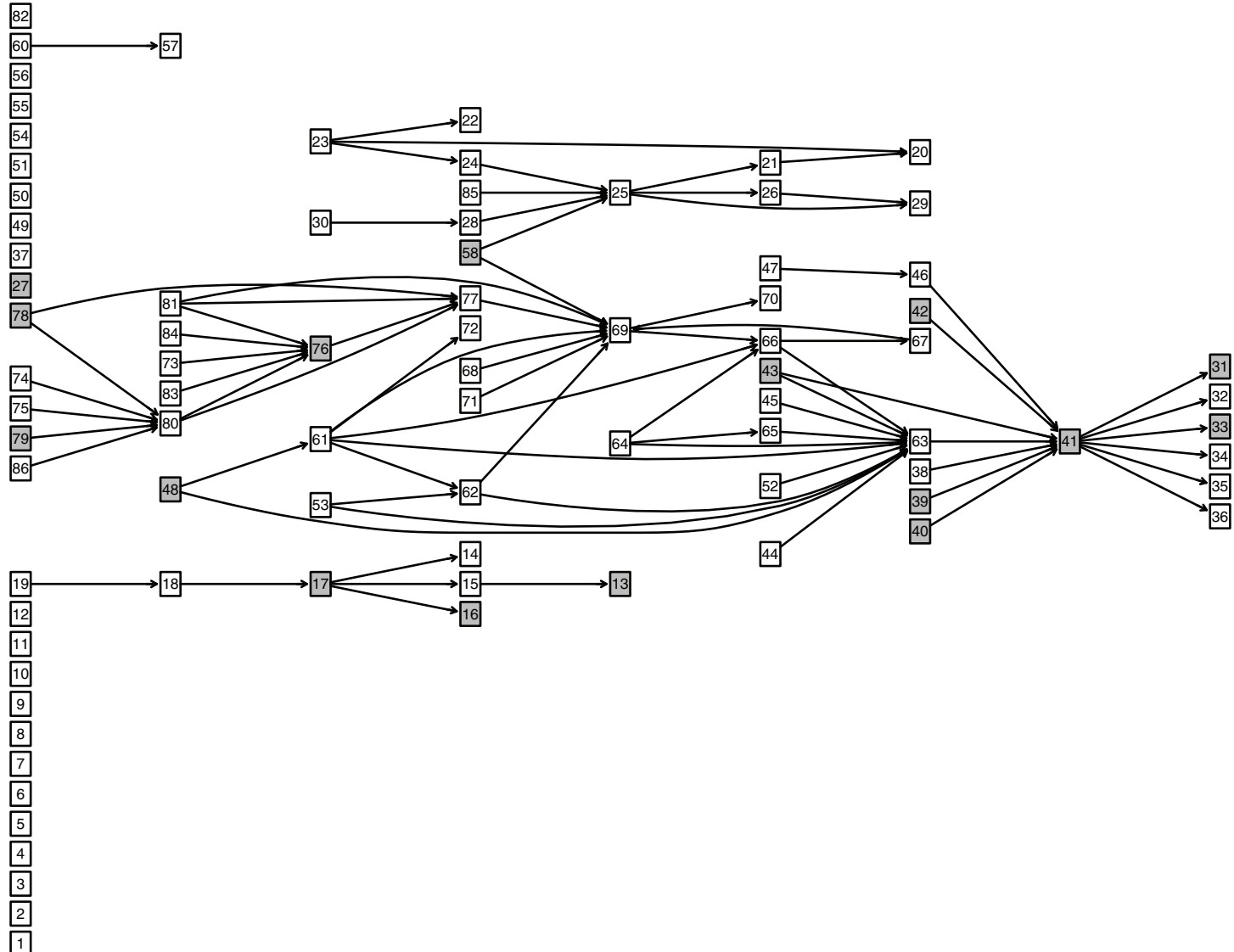

**Figure 10 The distribution of DEGs in Wnt signaling pathway using colorectal cancer datasets (GSE4183).** The nodes with grey color mean that these nodes contain DEGs; the nodes with white color mean that these nodes do not contain DEGs.

functional attributes in the signaling pathway, deepening our understanding of disease cells' biological behaviors. Additionally, SPFA used the effector genes with high signal variations to find the abnormal part of the disease-related pathway.

However, SPFA was weaker than MRGSE, Fisher, and PADOG when comparing the median $p$-values of target pathways. We assume this is due to the statistical models used. The probability $P_{sd}$ is evaluated by gene IDs permutation. Correlation differences are sometimes used to establish differential co-expression networks. This indicates that high correlation differences may exist in randomly-selected paired genes. The $p$-values may

increase when paired genes with high correlation differences are randomly selected. Future studies should use a better statistical model to resolve this problem. Additionally, the 33 gene expression datasets used in this work were still limited. More experiments need to be conducted to further validate SPFA's performance. A large number of normal and disease samples are also needed to locate the effector genes with high signal variations in disease-related pathways. These genes could then serve as effective module biomarkers for accurately detecting or diagnosing complex diseases, or as drug discovery targets. SPFA depends on manually-curated signaling pathways which play a small role in complex cellular progression. More signaling pathways need to be discovered for SPFA's optimal performance.

## CONCLUSIONS

In this study, we developed a new signaling pathway analysis method called SPFA. We compared this method's ability to identify altered signaling pathways against the other seven methods. SPFA showed better results than the seven other methods. Our results also showed that the SPFA method could help identify abnormal functional attributes under normal and disease conditions and the abnormal parts of a pathway during the disease biological process.

### Funding

This work was supported by grants from the National Natural Science Foundation of China (61871121, 61271055, and 61471112). The funders had no role in study design, data collection and analysis, decision to publish, or preparation of the manuscript.

### Grant Disclosures

The following grant information was disclosed by the authors:
National Natural Science Foundation of China: 61871121, 61271055, and 61471112.

### Competing Interests

The authors declare that they have no competing interests.

### Author Contributions

- Zhenshen Bao conceived and designed the experiments, performed the experiments, analyzed the data, authored or reviewed drafts of the paper, and approved the final draft.
- Bing Zhang performed the experiments, authored or reviewed drafts of the paper, and approved the final draft.
- Li Li analyzed the data, prepared figures and/or tables, authored or reviewed drafts of the paper, and approved the final draft.
- Qinyu Ge analyzed the data, prepared figures and/or tables, and approved the final draft.
- Wanjun Gu analyzed the data, prepared figures and/or tables, and approved the final draft.
- Yunfei Bai conceived and designed the experiments, analyzed the data, prepared figures and/or tables, authored or reviewed drafts of the paper, and approved the final draft.

## Data Availability

The R code of SPFA can be accessed at GitHub: https://github.com/ZhenshenBao/SPFA. The accession numbers of the 33 datasets used are available in Table 1.

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
