# Peer review of "Identifying disease-associated signaling pathways through a novel effector gene analysis"

_PeerJ, doi:10.7717/peerj.9695_

## Round 0.1 · original submission · Major Revisions

It would be extremely important to address in a robust manner the comments of both reviewers. Many of the same issues are raised by both, and they differ primarily in their view of how serious these issues are. Regardless, fundamental issues such as a clear statement of where this effort sits within the large context of this research area and sound mathematical measures of performance need to be addressed.

Reviewer 1 ·

Basic reporting

The article can be refined by improving few specific portions (mentioned in my detailed comments in general comments section) with regards to english and literature references.
Context and background was adequately provided.

Experimental design

This article introduces a new approach to signal pathway analysis named "Signaling Pathway Functional Attributes Analysis" (SPFA). It addresses the shortcomings of the previous analysis methods by incorporating functional attribute finding and dissecting which parts of the pathway are abnormal.
Research questions, aims and scope were well defined. Methods had adequate details and the code was also provided via Github. The approach investigated here was compared with 7 other approaches in the literature.

Validity of the findings

Conclusions are well stated and can be further enhanced by addressing the following questions in the general comments section.

Additional comments

Must address:
1. What are the reasons to choose the 3 mentioned rules to select the 33 data sets from KEGGdzPathwaysGEO and KEGGandMetacoreDzPathwaysGEO R-package. The authors should elaborate and justify the reasons for their 3 rules.
2. Follow up to the same, what would the impact on the analysis if the related target pathways of these datasets are not KEGG pathways?
3. How is the p-value combined to acquire the significance of each signaling pathway?
4. What was the rationale for choosing the 3 measures to compare all the signaling pathways analysis methods? Are there any other measures in literature which are used?


Good to address/incorporate (this should be treated as suggestions to enhance/improve this study or follow-up study):
1. Line 282: “signal variations received by these effector genes were also in the top ranking.” What was the exact rankings?
2. What factors contribute to the low sensitivity of SPFA compared to other signaling pathways compared in this manuscript? Is it the case where the statistical model presented here as high accuracy (prioritization) but low sensitivity?
3. Can one create a hybrid metric that would optimize all 3: sensitivity, specificity and prioritization instead of just summing the ranks from individual ones?
4. Are there certain target pathways for which SPFA performs better/worse than others? The performance distribution of SPFA within different target pathways would be nice to look at?



Minor english/grammatical modifications:
1. “contribute to disease cell biological process” is repetitive in line 23 and 24. The sentence should be re-written.
2. The line 138 should be removed: Add your materials and methods here.
3. Equation 1 dse is not defined, should be defined right after the equation.
4. Line 170: the year of the referenced article not mentioned: Jung & Berger, as the format of the reference followed in this manuscript is name and year. The same thing should be checked for all the references.
5. Equation 2: ASD acronym full form not mentioned in text. Does it stands for Accumulated signal deviation?
6. Equation 6 and 7: all the variables should be defined properly

·

Basic reporting

The writing is extremely poor, almost unreadable. Nearly every sentence requires revision. This includes the abstract. Only after reading the paper am I able to understand the abstract.

Some background is provided however very little about recent developments. Most references are pre-2014. Beyond that, there is one from 2019, two from 2016 and one from 2015. A quick pubmed search uncovers several more recent publications that describe new methods. This paper should at least mention them and state why they are not evaluated in the current publication. In fact, the 2016 publication that is cited proposes and evaluates a new method LEGO, however this method is not mentioned in the current manuscript. Their new method is compared to several other methods, but not rationale is provided for how they were selected for comparison or why others were omitted.

Structure is acceptable, however the authors appear to misunderstand the purpose of each section.

The manuscript is self-contained.

Experimental design

The article is in the Biological and Health Sciences.

The research aims to describe and evaluate a new method for identifying biological pathways whose activity differs between cases and controls. The new method attempts to incorporate signalling pathway structure into its search for affected pathways. Most previous methods ignore structure. There are other methods that do consider structure but these are not considered or discussed in much detail.

Evaluation of the new method is based on the approach of a previous publication (Tarca et al. 2013) although the authors mistakenly credit Bayerlová et al. (2015). Some 'cherry picking' is evident when they compare methods according to three measures, sensitivity, prioritization, and specificity. They correctly note that no method is likely to be best under all measures and choose to sum performance ranks for sensitivity and prioritization but not specificity. Under this ranking scheme, SPFA performs best. They fail to note in the text, however, that it is actually tied with PADOG (Table 2). Furthermore, if performance ranks are summed across all three measures, then PADOG actually turns out to be the overall winner (rank sum=7 compared to SPFA rank sum = 10). This is not mentioned in the text.

SPFA identifies dysregulation in pathways where correlation between pairs of genes with signaling relationships differs between cases and controls. Given this, it seems odd that the authors chose to use such small datasets to showcase their method (one 15v8 and the other 4v4). Does it even make sense to compare correlations calculated for n=4 samples and compare them to correlations calculated for another n=4 samples? Surely much larger datasets are available from GEO or certainly through TCGA.

The statistical significances of correlation differences in SPFA are evaluated by gene label permutation. Unfortunately this approach is likely to inflate statistical significance because genes within the same pathway tend to be correlated much more highly than genes in different pathways. Consequently, under random gene label permutation, correlations will tend to be much lower. It would be much better to compute a null distribution, as in Tarca et al. 2013, by permuting group labels instead.

Methods are sufficiently described. The authors provide code (https://github.com/ZhenshenBao/SPFA) however this lacks documentation and provides incomplete description of an example. Repeating the example requires the user to write additional code. The authors have clearly performed the complete analysis so it's not clear why the complete code is not provided. The R code relies on a few R packages: KEGGgraph, igraph, and ggplot. These dependencies are not documented.

Validity of the findings

The manuscript, including the abstract, claims that SPFA, unlike previous methods, is alone able "to help to find which functional attribute is in case of an exception between two conditions (normal vs. disease) and which parts of a pathway are abnormal in the disease biological process." This is simply not true. It is possible for example for any other method to evaluate enrichment in a sub-pathway.

That said, SPFA does appear to be different than previous methods in that it takes into account correlations between genes with signaling relationships in pathways. Because of this, it can potentially identify dysregulated pathways missed by methods focused on gene set enrichment.

Additional comments

Lacking from the manuscript is a more in-depth discussion that attempts to explain why the different methods perform the way they do. For example, PADOG likely has improved specificity because it down-weights the contribution of genes involved in multiple pathways. Given the superior performance of PADOG, perhaps a similar approach could be applied to improve SPFA specificity.

SPFA attempts to combine two different approaches based on enrichment expression differences and correlation differences. I'm curious why the correlation difference approach was not evaluated as an approach on its own since it identifies a different type of dysregulation. In fact, it might be more useful as part of a 2-stage approach to pathway analysis where enrichment is used to identify pathways and then correlation-differences are used to identify the most dysregulated relationships in that pathway.

Throughout it is assumed that the enrichment approach is entirely independent from correlation differences approach (hence the method for combining p-values on Line 210). However, this is not entirely obvious and should be more formally investigated. This outcome of this investigation is critical for a thorough understanding of SPFA.

It should be mentioned in the discussion that a weakness of the approach is its dependence on manually curated signaling pathways. The level of signaling knowledge varies significantly between and even within pathways.

Minor comments:
Line 175 ‘Euclidean distance’ should be ‘Network distance’.
Line 226 ‘Absolute difference’ implies that differences are always non-negative. However, the formulas given earlier suggest actual difference.
Line 246 Who is Michaela?

---

## Round 0.2 · Minor Revisions

Sorry for the long time taken for the review. The reviewer is now happy with the revision, which I also concur with. So we will generally accept your manuscript after you make some minor revision. Specifically, the term "disease cell behavior" is not standard and very difficult to comprehend without reading the manuscript. I will suggest changing the title to potentially attract more readership. An example could be "Identifying disease-associated signaling pathway through a novel effector gene analysis". The language may also need to be further polished. I found there were still typos and errors.

Reviewer 1 ·

Basic reporting

The authors significantly improved the manuscript in terms of english. The manuscript is much more understandable from its previous version.
In the present version ,the authors also did a good job in reviewing the current landscape in the introduction section compared to the previous version.

Experimental design

The authors also improved the method section and listed the reasons for collecting the data and using the comparison metrics. The authors also addresses the concerns I raised in the previous review.

Validity of the findings

Conclusions are well stated, and the authors also compared their results to the other 7 techniques and quoted those studies.

---

## Round 0.3 · accepted · Accept

Now the manuscript is suitable for publication. Congratulations!